# Laser-Plasma Spatiotemporal Cyanide Spectroscopy and Applications

**DOI:** 10.3390/molecules25030615

**Published:** 2020-01-31

**Authors:** Christian G. Parigger, Christopher M. Helstern, Benjamin S. Jordan, David M. Surmick, Robert Splinter

**Affiliations:** 1Physics and Astronomy Department, University of Tennessee, University of Tennessee Space Institute, Center for Laser Applications, 411 B.H. Goethert Parkway, Tullahoma, TN 37388, USA; chris.helstern@gmail.com; 2Nuclear Engineering Department, University of Tennessee, Tickle College of Engineering, 1412 Circle Drive, Knoxville, TN 37912, USA; bjordan1@vols.utk.edu; 3Physics and Applied Physics Department, University of Massachusetts Lowell, Lowell, MA 01854, USA; David_Surmick@uml.edu; 4Wellinq Medical, Van der Waals Park 22, 9351 VC Leek, The Netherlands; rsplinter@gmail.com

**Keywords:** plasma diagnostics, molecular spectra, diatomic molecules, plasma spectroscopy, laser spectroscopy, laser-induced breakdown spectroscopy, optical emission spectroscopy, hypersonic expansion

## Abstract

This article reports new measurements of laser-induced plasma hypersonic expansion measurements of diatomic molecular cyanide (CN). Focused, high-peak-power 1064 nm Q-switched radiation of the order of 1 TW/cm${}^{2}$ generated optical breakdown plasma in a cell containing a 1:1 molar gas mixture of N${}_{2}$ and CO${}_{2}$ at a fixed pressure of 1.1 × 10${}^{5}$ Pascal and in a 100 mL/min flow of the mixture. Line-of-sight (LOS) analysis of recorded molecular spectra indicated the outgoing shockwave at expansion speeds well in excess of Mach 5. Spectra of atomic carbon confirmed increased electron density near the shockwave, and, equally, molecular CN spectra revealed higher excitation temperature near the shockwave. Results were consistent with corresponding high-speed shadowgraphs obtained by visualization with an effective shutter speed of 5 nanoseconds. In addition, LOS analysis and the application of integral inversion techniques allow inferences about the spatiotemporal plasma distribution.

## 1. Introduction

Molecular-recombination spectra are readily measured following optical breakdown in gases. The cyanide (CN) molecule can be recognized rather early for time delays of the order of 100 ns after cessation of the laser pulse [1]. Gas mixtures are selected for the purpose of measuring CN spatiotemporal distributions associated with high-speed, hypersonic gas-dynamic expansion. The main goals and objectives of this work comprise (a) exploring CN measurement using optical emission spectroscopy subsequent to optical breakdown; (b) determining the effects of laser-induced shockwave for time delays of the order of 1 μs; (c) evaluating the spatial distribution of CN signals; (d) inferring CN temperature distribution; and (e) associating recorded shadowgraphs with laser-spectroscopy results. Applications of cyanide spectroscopy include various scientific areas, including medical research and selected engineering endeavors. In this work, formation of CN molecules occurs through recombination subsequent to laser-induced optical breakdown. Laser plasma is studied by measuring spatial distribution along the optical axis for selected time delays. For the mixture and flowing-gas experiments, chemical composition was governed by free but shockwave-affected expansion associated with laser-induced plasmas in gases.

Cyanides are connected to many living organisms, and are created through artificial processes, but they are highly toxic. Human exposure can occur through accidental ingestion or inhalation of cyanides. Synthetic materials can be found in 21st century buildings and furnishings that produce toxic-combustion byproducts like hydrogen cyanide (HCN) when burned [2]. Victims of these fires can have significant intake of HCN, which can lead to cyanide poisoning and ultimately death. Cyanide detection can identify cyanide-poisoning occurrences in humans and can assist in the determination of adequate treatment. Additionally, HCN can be produced by bacteria such as *Pseudomonas aeruginosa* [3]. Immune-compromised patients, such as those with cystic fibrosis (CF), are susceptible to increased morbidity and mortality when afflicted by these bacteria [3]. Detection of bacterial-produced HCN in the airways leads to earlier treatment of immune-compromised infected with these bacteria.

Diatomic molecule CN has remained a target for improved detection capability and deeper understanding, largely because engineers cannot replicate its effectiveness for certain industrial applications with any other compound [4]. For example, 90% of global gold-mining operations utilize CN because of its availability, effectiveness, and cost efficiency [5]. Although accidents, injuries, and environmental mishaps are possible, when they occur, they require specialized detection capability to ensure worker safety during remediation.

Engineering applications associated with plastic manufacturing that continue to use cyanide create a downstream problem for firefighters, analogous to that of fire victims mentioned above. Firefighters must go into buildings where plastics that may have used cyanide during their manufacture then evolve their constituents into the air as they are consumed by fire [6]. Again, detection and assessment of the amount of CN present is needed to help keep firefighters safe.

Engineers continue to look for means of obtaining CN properties that customers need but that do not have the potential health hazards that are present with CN [7,8]. Engineers have been seeking ways to lower cyanide utilization in the nitrile chemical industry [9]. Alternatives to cyanide are also sought for the mining industry, but this task has proven difficult because replacement candidates are often not as broadly applicable and must be tailored to the specific ore [10].

In addition, CN is a component of vehicle exhaust [11,12,13,14,15]. It is important for engineers to continue to research the details of CN detection and evaluation to be able to characterize the relative emission impact that different vehicles contribute [16]. There is CN production potential in other combustion processes as well, including the CN spectroscopy of trinitrotoluene (TNT) simulant molecules [17], and continued understanding of what level of CN production should be expected continues to drive interest and investigation into CN, even if engineers were able to replace CN in many industrial applications [18].

## 2. Experiment Details

The standard experiment components were used for laser-induced breakdown spectroscopy, and were summarized previously, e.g., see [1], but were included for completeness. The experiment arrangement consisted of a set of typical components for time-resolved laser-induced optical emission spectroscopy, or nanosecond laser-induced breakdown spectroscopy (LIBS) [19]. Primary instrumentation includes a Q-switched neodymium-doped yttrium aluminum garnet, Nd:Y${}_{3}$Al${}_{5}$O${}_{1}2$ (Nd:YAG) device (Quantel model Q-smart 850, USA) operated at the fundamental wavelength of 1064 nm to produce full-width-at-half-maximum 6 ns laser radiation with an energy of 850 mJ per pulse, a laboratory-type Czerny–Turner spectrometer (Jobin Yvon model HR 640, Fr) with 0.64 m focal length, equipped with a 1200 groove/mm grating, an intensified charge coupled device (Andor Technology model iStar DH334T-25U-03, USA) for recording temporally and spatially resolved spectral data, a laboratory chamber or cell with inlet and outlet ports, and a vacuum system, electronic components for synchronization, and various optical elements for beam shaping, steering, and focusing.

For the generation of optical-breakdown microplasma, a singlet lens (Thorlabs model LA1509-C, USA) was used close to the top entrance window of one arm of the chamber containing the 1:1 CO${}_{2}$:N${}_{2}$ atmospheric molar gas mixture (Airgas ultrahigh-purity N${}_{2}$ and research-grade CO${}_{2}$). For 1:1 imaging of the plasma onto the 100 μm spectrometer slit, a fused silica planoconvex lens (Thorlabs model LA4545, USA) was employed. For the CN experiments reported here, laser-pulse energy was attenuated with beamsplitters and apertures from 850 to 170 mJ/pulse. Measurements were performed with and without an Order Sorting Filter (Oriel model 51250, USA), with a cut-on wavelength of 309 nm and transmittance range of 325 nm, to evaluate the C I 193.09 nm atomic carbon line in second-order interference as identified in previous work [1].

In previous experiments, captured shadowgraphs of the breakdown plasma [20] served the purpose of visualizing the plasma expansion when using 850 mJ, 6 ns radiation. However, it is important to obtain shadowgraphs for plasma-excitation energies that were employed for time-resolved spectroscopy. Shadowgraphs reported in this work were captured using two separate laser devices (Continuum Surelite model SL I-10, USA) that could be externally operated to deliver laser pulses with a well-defined time delay showing less than ±1 ns trigger-jitter between the pulses. For visualization studies, both lasers were frequency-doubled to operate at the 2nd harmonic, 532 nm wavelength, and both beams were spatially overlapped. Both pulses could be delivered with a minimal time delay of 300 ns. Shadowgraphs were recorded by external synchronization of the Surelite and Quantel laser devices, and by externally triggering the camera (Silicon Video 9C10 color camera, USA) that recorded the images that were projected onto a screen.

## 3. Results and Discussion

### 3.1. Shadowgraphs

Two typical images are reported here. First, Figure 1 displays a shadowgraph of the IR 1064 nm laser-induced plasma for an excitation energy of 170 mJ 6 ns pulse width. The shadowgraph was captured using a single 5 ns pulse-width 532 nm beam.

The expanding shockwave was clearly visible, including the central plasma kernel. The image was nearly spherical; previously reported shadowgraphs that utilized 850 mJ excitation would reveal stronger deviation from spherical symmetry than the images obtained with a 170 mJ/pulse. The expanding shockwave radius was consistent with previous assessments of hypersonic laser-plasma expansion images [20] that employed Taylor–Sedov energy${}^{1/5}$ dependency [21],
(1)$$R\left(\tau \right)={\left(\frac{E}{\rho }\;\tau^{2}\right)}^{1/5}$$
of shockwave radius R(τ) as function of absorbed pulse energy *E*, gas density ρ, and time delay τ. Table 1 and Table 2 compare for energies of 160 and 200 mJ, respectively, computed blast-wave radii for standard ambient temperature and pressure (SATP) air, and the CN mixture. Consequently, measured shadowgraphs in air provided an excellent guide for the gaseous mixture.

Figure 2 exhibits a composite shadowgraph obtained by initiating laser plasma with the first, green, 90 mJ/pulse, 5 ns beam. Subsequently, initiating laser plasma at the shockwave with the second, green, 110 mJ/pulse, 5 ns beam spatially overlapped exactly, but was time-delayed by 0.45 μs. The composite image was obtained with the 0.45 μs time delay and an extra 30 ns optical time delay for each beam. The initial, elongated multiple breakdown in the central region was imaged with the first 5 ns pulse-width beam. The near-spherical shockwave was imaged with the second 5 ns pulse-width beam and the corresponding 30 ns time delayed beam. However, the second beam initiated breakdown at the forward side (left in image) and at the side of the incoming beam (right in image), indicated by a dark ring. The extent of that dark ring was due to the use of a 5 ns pulse-width beam, time-delayed by 30 ns for shadowgraph capture of expansion dynamics. The expansion speed of the second optical breakdown at the right of the image was well over one order of magnitude larger than hypersonic speed in SATP air.

Subsidiary measurement utilized a pyroelectric laser-pulse energy meter (Scientech model P50, USA) for determination of absorbed energy for the laser-induced plasma studies. Transmitted energy per pulse $E_{t}$ amounted to 20 ± 3 mJ for IR laser-induced plasma studies. This energy/pulse represents the optical-breakdown threshold. The energy/pulse threshold corresponded to the peak-power per area or irradiance threshold for SATP air optical breakdown [22]. For example, for perfect Gaussian beam focusing [23,24] with f/20, i.e., F/D=20 with *F* and *D* denoting the focal length and the beam diameter at the lens, respectively, for 1064 nm IR wavelength, λ, and from the beam radius, $w_{0}$,
(2)$$w_{0}=\frac{2}{\pi }\;\lambda \frac{F}{D}$$
one finds the focal area of $5.8\times 10^{-6}$ cm${}^{2}$. Using for $\Delta \tau_{\mathrm{pulse}}=6$ ns pulses the peak-power, $\frac{1}{2}E_{t}/\Delta \tau_{\mathrm{pulse}}$, of 1.7 MW (mean-field top-hat model for the temporal pulse distribution) one would find the irradiance threshold of $2.9\times 10^{11}$ GW/cm${}^{2}$. In view of Equation (Equation 1) for determination of shockwave radius, the absorbed energy per pulse should be used. However, the 1/5 exponent implied diminished dependency on the energy/pulse when in the range of 160 and 200 mJ, as displayed in Table 1 and Table 2. In other words,when considering effects from 220 and 200 mJ per pulse laser-plasma generations, a ratio of 220/200=1.1 or a 10% difference in energy/pulse would result in a 2% difference in the predicted shockwave radius for the same density and time delay.

### 3.2. Cyanide Spectra

The experiment series for the measurement of CN molecular distribution after optical breakdown includes evacuating the cell to a nominal backing pump pressure of the order of 1 Pa (10${}^{-2}$ Torr), followed by generating the mixture from ultrahigh pure N${}_{2}$ and research-grade CO${}_{2}$. Optical breakdown was generated inside the chamber at a rate of 10 Hz, with the laser beam focused with f/5 optics from the top, or parallel to the slit. The detector pixels were binned in four-pixel tracks along the slit direction, resulting in obtaining 256 spectra for each time delay. Recording of measurements with and without the Order-Sorting filter consist of 100 accumulations collected for 21 time delays at 250 ns steps. These runs allowed for the evaluation of CN plasma spatial and temporal characteristics, and the C I 193.09 nm atomic carbon line in second order. Figure 3 and Figure 4 illustrate spatiotemporal spectra that were recorded along the line of sight and were accumulated over 100 individual laser-plasma events. The slit height corresponds to the z-direction, and the line of sight corresponds to the y-direction in reference to Figure 1 and Figure 2.

### 3.3. Cyanide Temperature

Both line-of-sight and Abel-inverted datasets of the CN spectra were subjected to analysis with the Nelder–Mead temperature (NMT) program that uses the Nelder–Mead algorithm [25] and was made available along with the required CN violet system dataset [26,27]. The fitting utilized the NMT program and CN line-strength data; both are available in [26]. The theoretical background and development of the line-strength data are found in [27]. Figure 5 illustrates a typical CN spectrum and fit of line-of-sight data accomplished in this work.

Figure 6 exhibits analysis results of the filtered, line-of-sight molecular CN spectra. The figures reveal the occurrence of the outgoing shockwave, along with temperature variations in the central region of the plasma. These data of slit height vs. temperature show increased temperature near the edges.

### 3.4. Electron Density

Unfiltered spectra contained an overlap of the C I 193.09 nm atomic carbon line in second order and the 2-2 CN band head of 386.19 nm, where filtered spectra only contained the 2-2 CN band head of 386.19 nm. C I 193.09 nm atomic carbon line in second-order Stark widths were evaluated for filtered and unfiltered spectra using peak-fitting programs. The difference between the filtered and unfiltered spectra was obtained by applying deconvolution. Figure 7 illustrates the results for the widths at full width half maximum of the carbon peak. Figure 7 reveals that larger Stark widths were seen near the edges of the plasma, while smaller Stark widths were near the center of the plasma.

Electron number density $n_{e}$ can be determined from the Stark full width at half maximum, $\Delta \lambda_{\mathrm{Stark}}$, of the C I 193.09 nm atomic carbon line [28], measured in second order,
(3)$$\Delta \lambda_{\mathrm{Stark}}\left(\mathrm{nm}\right)=2\;w\left(\mathrm{nm}\right)\;n_{e}\left(10^{17}{\mathrm{cm}}^{-3}\right)$$
where width parameter, *w* was extrapolated [28,29] to amount to w≈0.0029 nm. Figure 8 displays the calculated electron densities versus slit height. The calculated electron densities are of the order of $n_{e}≃10^{17}{\mathrm{cm}}^{-3}$ in the central region. Higher electron densities were seen near the edges of the plasma, while smaller electron densities were near the center of the plasma.

Using the Taylor–Sedov blast-wave model (see Equation (Equation 1)), plasma radius $R_{\mathrm{plasma}}$, at time delay of 450 ns, was ≈1.9 mm, which would indicate higher electron density near the shockwave, which is consistent with results shown in Figure 8.

### 3.5. Cyanide Spectra in Flowing Gas

The experiment series for the measurement of CN molecular distribution after optical breakdown for the flowing-gas mixture utilized a similar process as discussed in the previous sections. The experiments included supplying the cell with a constant flow of ultrahigh pure N${}_{2}$ and research-grade CO${}_{2}$ gas mixture instead of a fixed volume of the mixture. The flow rate of the mixture entering and leaving the cell was 100 mL/min and was monitored using a flowmeter (Cole-Parmer Instr. Co. FM 112-02ST, USA). Optical breakdown was generated inside the chamber at a rate of 10 Hz, with the laser beam focused with f/5 optics from the top or parallel to the slit. The detector pixels were binned in four-pixel tracks along the slit direction, resulting in obtaining 256 spectra for each time delay. Recording of measurements with and without the Order-Sorting filter consist of 100 accumulations collected for 21 delays at 250 ns steps.

Applying the NMT program to the filtered, line-of-sight molecular CN-spectra produced in the flowing-gas mixture yielded the results shown in Figure 9. The figure reveals the occurrence of the outgoing shockwave along with temperature variations in the central region of the plasma, and these slight-height vs. temperature data showed increased temperatures near the edges, analogous to the results with the fixed amount of gas mixture. Comparing these results to Figure 7, calculated CN temperatures for a fixed amount of gas mixture, it appears that the highest calculated CN temperatures for Figure 7 were towards the bottom of the plasma, where the highest calculated CN temperatures for the flowing gas were towards the top of the plasma.

Just as in the fixed gas mixture, unfiltered spectra contained an overlap of the C I 193.09 nm atomic carbon line in second order and the 2-2 CN band head of 386.19 nm, where filtered spectra only contained the 2-2 CN band head of 386.19 nm. The C I 193.09 nm atomic carbon line in second-order Stark widths was evaluated for filtered and unfiltered spectra using peak-fitting programs, and the difference between filtered and unfiltered spectra was obtained by applying deconvolution. Figure 10 illustrates the results for the width at full-width-half-maximum of the carbon peak, and reveal that larger Stark widths were seen near the edges of the plasma, while smaller Stark width were near the center of the plasma, similar to the results in the fixed gas mixture.

Using Equation (Equation 3), electron densities could be calculated for the flowing-gas mixture. Figure 11 displays the calculated electron densities versus slit height, and the electron densities were on the order of $≃10^{17}$ cm${}^{-3}$ in the central region. Higher electron densities were seen toward the edges of the plasma, and smaller electron densities were near the center of plasma, analogous to the results for the fixed gas mixture. Additionally, the plasma at 450 ns delay in Figure 11 displayed higher electron densities near the shockwave (at ≈1.9 mm), which is consistent with the Taylor–Sedov blast-wave model and the presented experiment results with a fixed gas mixture in the cell.

### 3.6. Abel-Inverted Spectra

In view of the shadowgraphs that were recorded in the 0.5 to 1 μs range (see Figure 1 and Figure 2), it would be acceptable to apply an Abel inverse transform. Deviation from spherical symmetry is minimal for time delays of the order of 1 μs provided laser-induced excitation is accomplished with nanosecond IR radiation of the order of 200 mJ/pulse or less. Analysis of the molecular CN spectra follows the same symmetrization methods as applied for atomic hydrogen spectra [30,31].

The use of Chebyshev polynomials for Abel inversion of integral equation
(4)$$I\left(z,\lambda \right)=2\int_{z}^{\rho }\varepsilon \left(r,\lambda \right)\frac{r}{\sqrt{r^{2}-z^{2}}}dr$$
allows one to directly invert measured data. The *z*-direction corresponds to slit height, and line-of-sight integration was along the y-direction in view of Figure 1. In this work, choosing the number of 15 polynomials for the inversion [32,33] maintained fidelity of the spectra and was equivalent to the use of a digital filter resulting in broadening computed radial spectra. A smaller number of polynomials would cause smaller spectral resolution. Measured line-of sight data I(z,λ) along slit dimension *z* were inverted for each wavelength λ to obtain radial intensity distribution ε(r,λ), with the upper limit much larger ($\rho \gg R_{\mathrm{plasma}}$) than radius, $R_{\mathrm{plasma}}$) of the plasma.

The radial spectra were computed from the two-dimensional line-of-sight data (see Figure 3 and Figure 4) that were accumulated with an intensified camera attached to the spectrometer, allowing one to resolve spectra along the slit. The recorded data were calibrated, corrected for system sensitivity using standard lamps, and prepared for Abel inversion. Usually, when recording 2D spectra, slight wavelength variations towards the edges of the slit occur, but these variations were taken care of in preparing the data for Abel inversion. Figure 12 and Figure 13 illustrate the computed Abel inverted spectra. Only subtle differences are recognizable near the 2-2 CN band edge due to the C I line when comparing Figure 12 and Figure 13; however, Figure 12 also contains electron-density information that confirmed higher electron density near the edges than the central regions, and, consequently, higher electron temperature for adiabatic expansion.

Analysis of Abel-inverted data was expected to reveal similar results for the shockwave, but in view of the shadowgraphs in Figure 1 and Figure 2, variations of CN distribution inside the shockwave and the plasma kernel were anticipated.

## 4. Conclusions

The main results of the work are summarized here: Primarily shockwave expansion affects the formation of CN molecules as the plasma expands. For time delays of the order of 1 μs, higher CN and electron concentrations occur near the shockwave than those in the central region of the plasma. The novelty of the laser-plasma experiments includes measurements of shockwave-expansion characteristics by tracing molecular CN formed by recombination. For time delays of the order of several μs after plasma generation, regular CN molecular distributions occur in the plasma, namely, hotter central than cooler peripheral plasma, as indicated by CN signals.

Laser-plasma expansion following optical breakdown occurs at well above hypersonic speed. Measurements of recombination CN spectra within the first few microseconds yielded results as expected from atomic hydrogen Balmer series laser spectroscopy: Analysis revealed higher electron and higher CN excitation temperature near the shockwave than in the central region for time delays of the order of one microsecond. The expansion characteristics were deduced from systematic analysis of the recorded line-of-sight spectra. Application of Abel inversion was reasonable and supported by shadowgraphs recorded in SATP air for laser-plasma initiation for similar energy/pulse, as utilized for measurements of molecular spectra.

Further development of CN diagnosis with laser spectroscopy should focus on accurate measurements of spatial distributions with simultaneous recording of spectra at different angles. This would allow the examination of deviation from spherical symmetry by invoking inverse Radon transforms, viz. by applying computed tomography. For the various applications mentioned in this work, measurements of spatial distribution as plasma expands would also alleviate averaging along the line of sight.

## Figures and Tables

**Figure 1 molecules-25-00615-f001:**
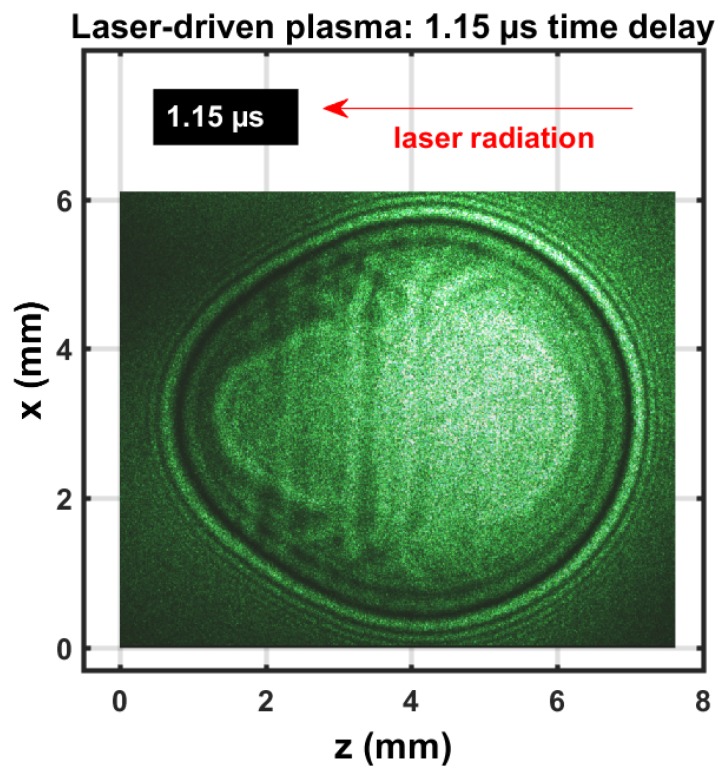
Single-shot shadowgraph of expanding laser-induced plasma initiated with a 170 mJ, 6’ns, 1064 nm focused beam, and imaged using a 5 ns, 532 nm backlight that was time-delayed by 1.15 ± 0.05 μs.

**Figure 2 molecules-25-00615-f002:**
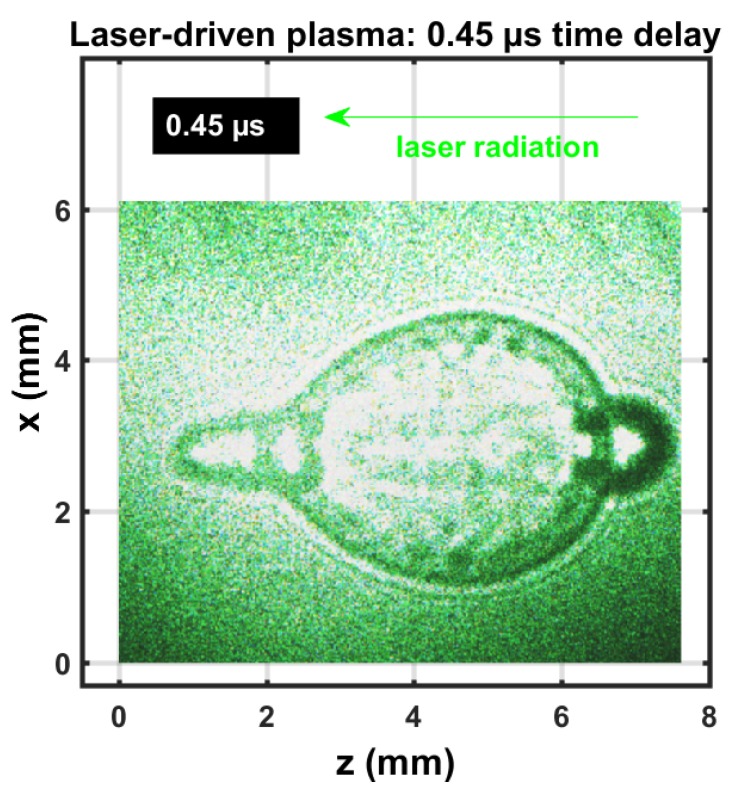
Single-shot shadowgraph of laser-induced plasma initiated with 90 and 110 mJ, 5 ns, 532 nm focused beams, and imaged using two 5 ns, 532 nm back lights that were time-delayed by 0.03, 0.45, and 0.48 μs.

**Figure 3 molecules-25-00615-f003:**
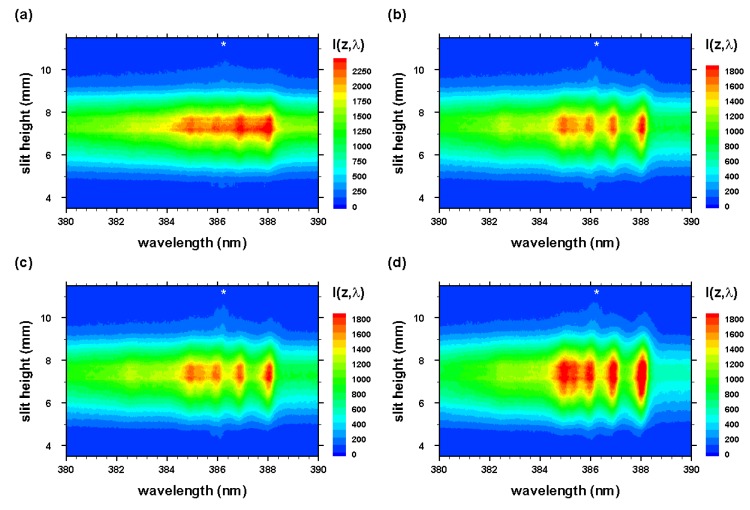
Optical breakdown CN spectra in a 1:1 molar CO${}_{2}$:N${}_{2}$ atmospheric gas mixture for time delays of (**a**) 200, (**b**) 450, (**c**) 700, and (**d**) 950 ns. Spectrometer-detector gate width: 125 ns. * , second-order neutral carbon line.

**Figure 4 molecules-25-00615-f004:**
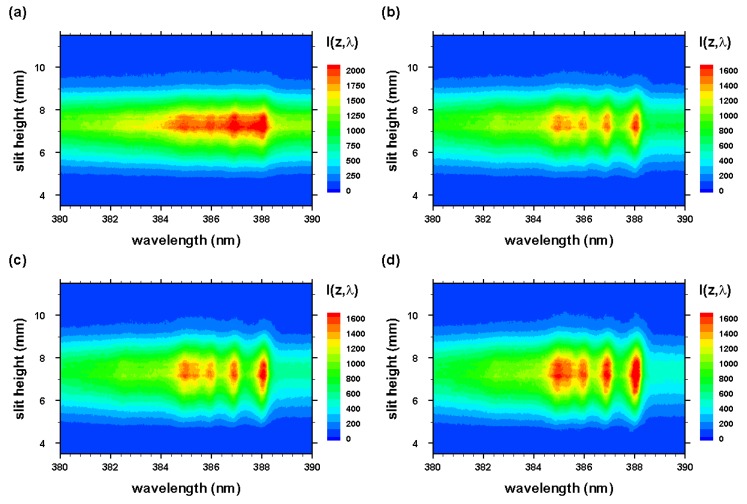
Optical breakdown CN spectra as in Figure 3 for time delays of (**a**) 200, (**b**) 450, (**c**) 700, and (**d**) 950 ns, recorded with 309 nm cut-on wavelength filter for suppression of 193.09 nm neutral carbon line contributions in second order.

**Figure 5 molecules-25-00615-f005:**
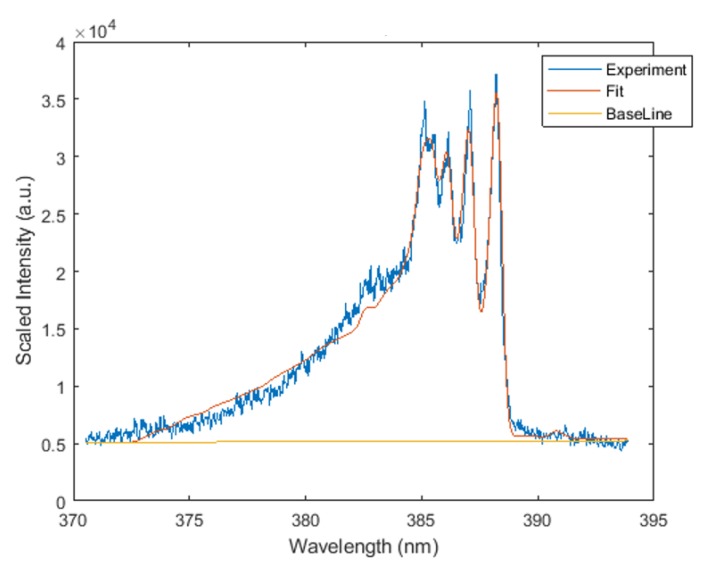
Typically fitted CN spectrum, T=8947 K, fit FWHM=0.47 nm.

**Figure 6 molecules-25-00615-f006:**
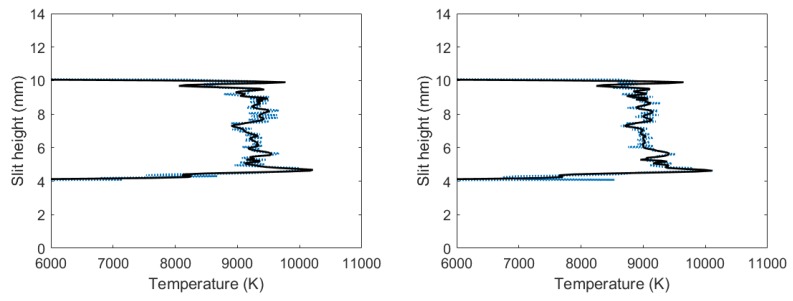
Temperature vs. slit height for filtered CN spectra with (**left**) 700 and (**right**) 950 ns time delay.

**Figure 7 molecules-25-00615-f007:**
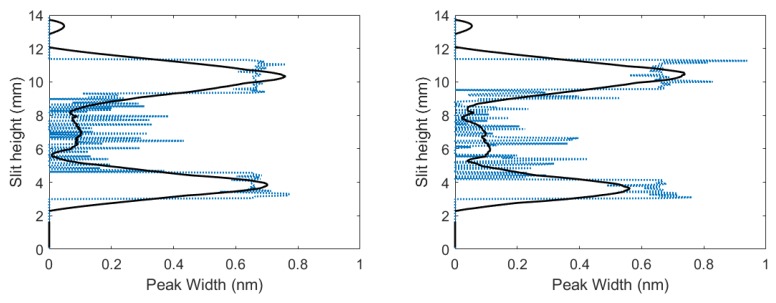
Inferred widths of C I 193.09 nm atomic carbon line in second order vs. slit height for time delays of (**left**) 700 and (**right**) 950 ns.

**Figure 8 molecules-25-00615-f008:**
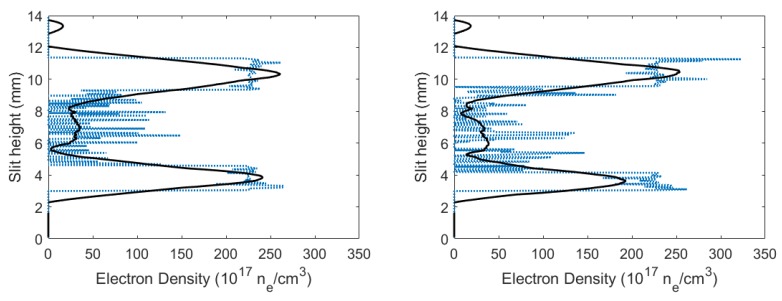
Calculated electron densities versus slit height for time delays of (**left**) 450 and (**right**) 950 ns.

**Figure 9 molecules-25-00615-f009:**
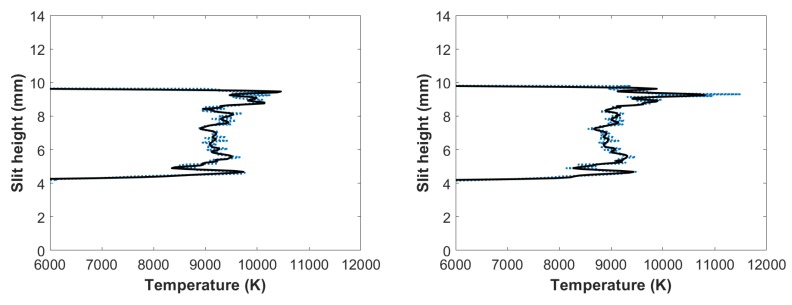
Temperature vs. slit height for flowing gas filtered CN spectra: (**left**) 700 and (**right**) 950 ns time delay.

**Figure 10 molecules-25-00615-f010:**
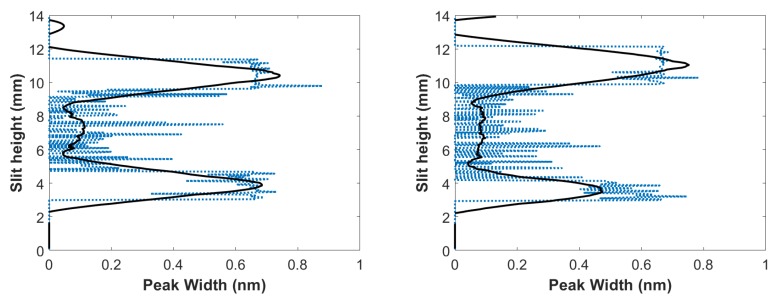
CN flowing-gas inferred widths of C I 193.09 nm atomic carbon line in second order vs. slit height for time delays of (**left**) 700 and (**right**) 950 ns.

**Figure 11 molecules-25-00615-f011:**
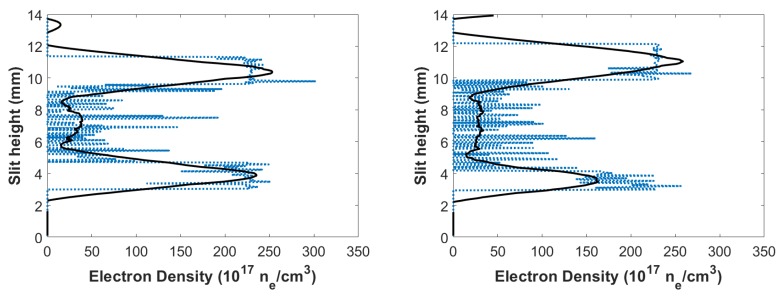
Calculated CN flowing-gas electron densities vs. slit height for time delays of (**left**) 450 and (**right**) 950 ns.

**Figure 12 molecules-25-00615-f012:**
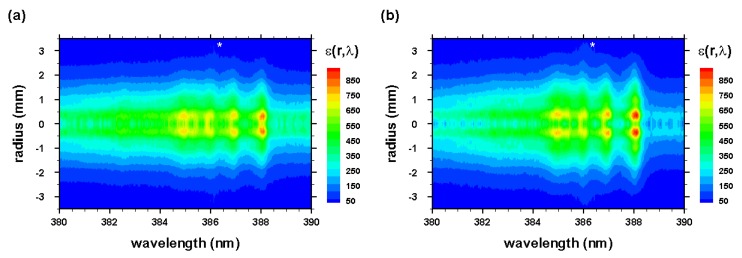
Abel-inverted data of Figure 3 for time delays of (**a**) 450 and (**b**) 950 ns. * , second-order neutral carbon line.

**Figure 13 molecules-25-00615-f013:**
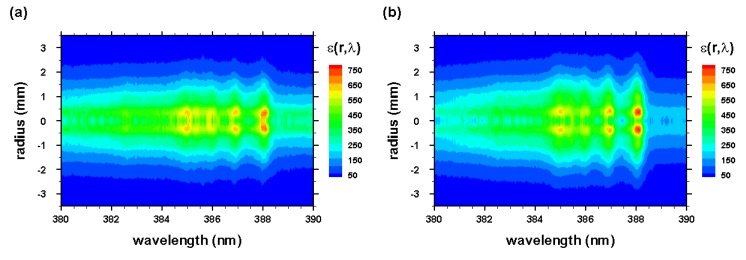
Abel-inverted data of Figure 4 for time delays of (**a**) 450 and (**b**) 950 ns.

**Table 1 molecules-25-00615-t001:** Computed shockwave radii for standard ambient temperature and pressure (SATP) air and for molar cyanide (CN) mixture, 160 mJ.

Time Delay (ns)	r (mm) for Air (ρ=1.2 kg/m${}^{3}$)	*R* (mm) for CN Mix (ρ=1.63 kg/m${}^{3}$)
200	1.40	1.31
450	1.93	1.82
700	2.31	2.17
950	2.61	2.45
1200	2.86	2.69
1450	3.09	2.90

**Table 2 molecules-25-00615-t002:** Computed shockwave radii for SATP air and molar CN mixture, 200 mJ.

Time Delay (ns)	r (mm) for Air (ρ=1.2 kg/m${}^{3}$)	*R* (mm) for CN Mix (ρ=1.63 kg/m${}^{3}$)
200	1.46	1.37
450	2.02	1.90
700	2.41	2.27
950	2.73	2.56
1200	2.99	2.81
1450	3.23	3.04

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
