# Peer review of "Laser-Plasma Spatiotemporal Cyanide Spectroscopy and Applications"

_molecules, 2020, doi:10.3390/molecules25030615_

Round 1
Reviewer 1 Report
This paper reports new measurements of laser-induced plasma hypersonic expansion of cyanide molecule (CN). The motivations, the method used as well as the results obtained are very clearly presented and discussed so I see no correction or improvement to suggest to the authors. I am therefore of the opinion that the manuscript can be accepted for publication in its present form.
Author Response
Reviewer 1
This paper reports new measurements of laser-induced plasma hypersonic expansion of cyanide molecule (CN). The motivations, the method used as well as the results obtained are very clearly presented and discussed so I see no correction or improvement to suggest to the authors. I am therefore of the opinion that the manuscript can be accepted for publication in its present form.
We appreciate the comments.
Minor edits are included as suggested by reviewers 2 and 3.
Reviewer 2 Report
The presented article contains the results of new measurements of laser-induced plasma hypersonic expansion of diatomic molecular cyanide. It was shown that molecular CN spectra reveal higher excitation temperature near the shockwave. The results are consistent with corresponding high-speed shadow graphs. Moreover, authors have used LOS analysis and application of integral inversion techniques which allow inferences about spatio-temporal distribution of the plasma. The results are obtained at high modern level and their correctness is beyond any doubt.
Nevertheless, I have some minimal but important remarks:
Authors should clarify the main goal and objectives of the study in the introduction It is necessary to indicate a chemical reaction leading to the formation of a CN molecule It should be explained how the plasma chemical composition was controlled In conclusion authors should briefly highlight the main results of the study and determine their novelty in comparison with existing data, if possible. Finally it is necessary to indicate new ways of further development of this field of scienceIn general, I believe that this interesting work can and should be published in the journal “Molecules” after minor revision.
Author Response
Reviewer 2
The edits in response to reviewer 2 are highlighted in green.
Comments and Suggestions for Authors
The presented article contains the results of new measurements of laser-induced plasma hypersonic expansion of diatomic molecular cyanide. It was shown that molecular CN spectra reveal higher excitation temperature near the shockwave. The results are consistent with corresponding high-speed shadow graphs. Moreover, authors have used LOS analysis and application of integral inversion techniques which allow inferences about spatio-temporal distribution of the plasma. The results are obtained at high modern level and their correctness is beyond any doubt.
Nevertheless, I have some minimal but important remarks:
Authors should clarify the main goal and objectives of the study in the introduction
First paragraph of introduction addition: The main goals and objectives of this work comprise (a) explore measurement of CN using optical emission spectroscopy subsequent to optical breakdown; (b) determine effects of the laser-induced shock wave for time delays of the order of 1 µs; (c) evaluate spatial distribution of CN signals; (d) infer CN temperature distribution; and (e) associate recorded shadow graphs with laser spectroscopy results. We also added 7 new references: Refs. [5],[7-8],[11-14]. See highlighted references in the References-section.
It is necessary to indicate a chemical reaction leading to the formation of a CN molecule.
We included towards the end of the first paragraph of introduction: In this work, formation of the CN molecule occurs through recombination that occurs subsequent to laser-induced optical breakdown.
It should be explained how the plasma chemical composition was controlled
We further included at the end of the introduction: The laser plasma is studied by measuring the spatial distribution along the optical axis for selected time delays. For the mixture and the flowing gas experiments, the chemical composition is governed by free but shock wave affected expansion associated with laser-induced plasmas in gases.
In conclusion authors should briefly highlight the main results of the study and determine their novelty in comparison with existing data, if possible.
Added the following paragraph in the conclusions: The main results of the work are summarized here: Primarily shock wave expansion affects the formation of CN molecules as the plasma expands. For time delays of the order of 1 µs, higher CN and electron concentrations occur near the shock wave than those in the central region of the plasma. Novelty of the laser plasma experiments include measurements of shock-wave expansion characteristics by tracing molecular CN formed by recombination. For time delays of the order of several µs after plasma generation, regular CN molecular distributions occur in the plasma, namely, hotter central than cooler peripheral plasma as indicated by CN signals.
Finally it is necessary to indicate new ways of further development of this field of science
Added the last paragraph in conclusions: Further development of CN diagnosis with laser-spectroscopy should be focused towards accurate measurements of the spatial distributions with simultaneous recording of spectra at different angles. This would allow examination of deviation from spherical symmetry by invoking inverse Radon transforms, viz. by applying computed tomography. For the various applications mentioned in this work, measurements of the spatial distribution as the plasma expands would also alleviate averaging along the line-of-sight.
In general, I believe that this interesting work can and should be published in the journal “Molecules” after minor revision.
Reviewer 3 Report
The paper by Christian et al. use laser-plasma spatio-temporal spectroscopy to measure the cyanide generated. The results are very interesting and useful for CN analysis. However, there remains some issues to be more clear. So I can’t recommend the acceptance for the current form till my concerns below have well been responded:
-Motivation is not clear. The authors listed a lot of reports to demonstrate the importance of the CN detection. But why do you use the proposed method still not clear? How it is compared with other solutions?
-Some abbreviations are used with out suitable definition. Please revise them.
-For Eq. (1), please check ‘R(t)’ or ‘R(τ)’, which are different.
-In line 113, ‘a ratio of 220/200=1.1’, how the values 220 comes from?
-Fig. 4 is recorded with 309 nm cut-on wavelength filter. But why do you use it? Any comparison and discussion with Fig. 3 and 4?
-In Fig. 5, it is not quite clear about the fitting theory (equation, function, etc.).
-The format of the references must be checked and unified.
Author Response
Reviewer 3
The edits in response to reviewer 3 are highlighted in blue.
The paper by Christian et al. use laser-plasma spatio-temporal spectroscopy to measure the cyanide generated. The results are very interesting and useful for CN analysis. However, there remains some issues to be more clear. So I can’t recommend the acceptance for the current form till my concerns below have well been responded:
-Motivation is not clear. The authors listed a lot of reports to demonstrate the importance of the CN detection. But why do you use the proposed method still not clear? How it is compared with other solutions?
-Some abbreviations are used with out suitable definition. Please revise them.
Added: trinitrotoluene (TNT) – see newline 59; neodymium-doped yttrium aluminum garnet, Nd:Y3Al5O12 (Nd:YAG) – see newline 68
-For Eq. (1), please check ‘R(t)’ or ‘R(τ)’, which are different.
Changed R(t) to R(τ)
-In line 113, ‘a ratio of 220/200=1.1’, how the values 220 comes from?
Added the text in front of ‘ a ratio of 220/200 ‘: when considering effects from 220 mJ and 200 mJ per pulse laser-plasma generation,
-Fig. 4 is recorded with 309 nm cut-on wavelength filter. But why do you use it? Any comparison and discussion with Fig. 3 and 4?
Added in caption for Fig.4: for suppression of 193.09-nm neutral carbon line contributions in 2nd order
-In Fig. 5, it is not quite clear about the fitting theory (equation, function, etc.).
Added a new Ref. and the text: that uses the Nelder-Mead algorithm [25] and
Further added: The fitting utilizes the NMT program and CN line-strangth data, both are available as a supplement in Reference [26]. The theory background and development of the line-strength data is communicated in Ref. [27].
-The format of the references must be checked and unified.
Checked and edited the references: included semicolons.
However, some newer references do not show page numbers, rather they show article reference numbers.